Variation in the pelvic and pectoral girdles of Australian Oligo–Miocene mekosuchine crocodiles with implications for locomotion and habitus

Stein Michael D. michael.stein@student.unsw.edu.au 1
Yates Adam 2
Hand Suzanne J. 1
Archer Michael 1
1 PANGEA Research Centre, School of Biological, Earth and Environmental Sciences, University of New South Wales , Sydney , New South Wales , Australia
2 Museum of Central Australia , Alice Springs , Northern Territory , Australia
Hutchinson John
Electronic publication date: 2017 Jun 30
Publication date: 2017
Volume: 5
Electronic Location ID: e3501
Received 2017 Mar 20; Accepted 2017 Jun 4
Copyright: ©2017 Stein et al.
Copyright year: 2017
Copyright holder: Stein et al.
License: This is an open access article distributed under the terms of the Creative Commons Attribution License, which permits unrestricted use, distribution, reproduction and adaptation in any medium and for any purpose provided that it is properly attributed. For attribution, the original author(s), title, publication source (PeerJ) and either DOI or URL of the article must be cited.
License URL: https://creativecommons.org/licenses/by/4.0/

Keywords: Crocodile, Evolution, Mekosuchinae, Erect high-walking, Paleoecology, Oligo–Miocene, Australia, Pelvic and pectoral girdles, Morphological radiation, Pillar-erect locomotion

Funding: PANGEA Research Centre (UNSW) Riversleigh Society Inc Australian Research Council LP100200486 DP130100197 This research was supported by postgraduate grants to MS from the PANGEA Research Centre (UNSW) and Riversleigh Society Inc., and Australian Research Council grants (LP100200486 and DP130100197) to MA and SJH The funders had no role in study design, data collection and analysis, decision to publish, or preparation of the manuscript.

==============================
Australian Oligo–Miocene mekosuchines (Crocodylia; Crocodyloidea) display wide diversity in cranial shape and inferred hunting strategies. Terrestrial habitus has been inferred for these distinctive predators. A direct morphological signal for locomotion can be expected in the postcrania, particularly the pelvic and pectoral girdles. Here we describe fossil materials of the girdles, which chart their morphological variation in the subfamily from Eocene through to Middle Miocene. Over this period, both girdles undergo significant morphological changes. Notably, an enclosed, ventrally orientated acetabulum in the ilium is developed in one lineage. This recapitulates the erect parasagittal configuration of the pelvic limb seen in many Mesozoic crocodylomorph lineages, suggesting consistent use of erect high-walking in these mekosuchines. Other pelves from the same Oligo–Miocene deposits display morphology closer to modern crocodilians, suggesting a partitioning of locomotory strategy among sympatric mekosuchines. Plesiomorphic and derived pelvic girdles are distinguishable by parsimony analysis, and the earliest examples of the mekosuchine pelvis more closely resemble gavialids and alligatorids while latter forms converge on crown group crocodylids in the morphology of the iliac crest. This suggests that a revaluation of the base relationship of Mekosuchinae within Eusuchia is necessary.

Introduction

A common perception of crocodilians is that they are morphologically conservative and in this sense modern crocodilians are sometimes thought of as ‘living fossils’. In fact, however, the fossil record documents that crocodylomorphs underwent a morphologically dynamic radiation following divergence of the Pseudosuchia (crocodile-line archosaurs) from the Avemetatarsalia (pterosaurs, dinosaurs and birds) 245–249 million years ago [Ma] (Markwick, 1998; Reilly & Elias, 1998; Brochu, 2003; Brusatte et al., 2008; Sereno & Larsson, 2009; O’Connor et al., 2010; Oaks, 2011; Stubbs et al., 2013; Toljagic & Butler, 2013; Mannion et al., 2015; Wilberg, 2017). Comparisons of the pelvic girdles in crocodilians, dinosaurs and birds have been the subject of several studies on the evolution of locomotion in these groups (Romer, 1923a; Romer, 1923b; Parrish, 1987; Gatesy, 1991; Meers, 2003; Hutchinson, 2006; Irmis, 2007; Abdala & Diogo, 2010; Schachner, Manning & Dodson, 2011; Chamero, Buscalioni & Marugan-Lobon, 2013). Current consensus holds that a semi-aquatic habitus was adopted sporadically among Mesozoic crocodylomorphs, and definitively in eusuchians by the Late Cretaceous (Parrish, 1987; Hutchinson, 2006). Like all aquatic amniotes, eusuchians returned to the water rather than emerged from it, a fact reflected in their morphology (Parrish, 1986; Parrish, 1987; Sereno, 1991; Reilly & Elias, 1998). Because lacustrine and fluvial environments are more conducive to preservation than most terrestrial habitats, there is a taphonomic bias towards semi-aquatic forms in the fossil record. This, coupled with the dominance of a semi-aquatic habitus in extant crocodilian faunas, has led to this common perception that crocodiles as a whole have maintained a relatively conservative body form, a perception that reflects the presumption that most have occupied more or less similar freshwater lactustrine/fluviatile habitats (Brochu, 2003). Throughout their history, however, there are also examples of purely terrestrial and marine crocodylomorphs (Colbert, 1946; Buckley et al., 2000; Schwarz, Frey & Martin, 2006; Sereno & Larsson, 2009; O’Connor et al., 2010; Sertich & Groenke, 2010; Pol et al., 2012; Puertolas-Pascual et al., 2016).

Australian mekosuchine crocodiles developed a wide variety of cranial morphologies indicative of varied hunting strategies in the Oligo–Miocene (Hecht & Archer, 1977; Willis, Murray & Megirian, 1990; Megirian, Murray & Willis, 1991; Willis & Molnar, 1991; Willis, 1993; Willis, Molnar & Scanlon, 1993; Megirian, 1994; Salisbury & Willis, 1996; Willis, 1997a; Willis, 1997b; Willis, 2001; Molnar, Worthy & Willis, 2002; Willis & Molnar, 2008; Stein, Hand & Archer, 2016). This includes indications of a more terrestrial lifestyle. Species of Quinkana, Mekosuchus and possibly Trilophosuchus display lateralization of orbits and nares (Willis, 1993; Megirian, 1994). Quinkana is further characterised by mediolateral compression of the rostrum and serrate labiolingually compressed (ziphodont) teeth, a trait associated more closely with active cursorial hunting than the ambush strategies ubiquitous in large-bodied extant crocodilians (Colbert, 1946; Megirian, Murray & Willis, 1991; Megirian, 1994; Pol et al., 2012).

A shift towards a more terrestrial habitus during the Oligo–Miocene should be reflected in anatomical features relating to locomotion such as in the limbs, pectoral and pelvic girdles. Previous work has examined the early mekosuchine forelimb (Stein et al., 2012). Fossil material of the pectoral and pelvic girdles has been recovered from sites across northern Australia. These specimens allow an overview of the evolution of locomotion of mekosuchines from the Eocene to Middle Miocene, with comparisons revealing significant morphological changes over this period. The present paper describes these changes and their implications for locomotory behaviours during the zenith of mekosuchine diversity.

Figure 1 Locality map.

Map showing referenced fossil localities noted in text. Scale bar equals 500 km. Map data © 2016 GBRMPA, Google.

Materials & Methods

Fossils representing the pectoral and pelvic girdles of mekosuchine crocodiles were collected from Cenozoic localities in both Queensland and the Northern Territory (Fig. 1). Queensland localities include Tingamarra near the township of Murgon, and The Narrows Graben near Gladstone, located in the southeast and central east of the state respectively, and the Riversleigh World Heritage Area located in the northwest. The Tingamarra Local Fauna (LF; sensu Archer et al., 1989) is regarded as Early Eocene in age (Godthelp et al., 1992), while The Narrows Graben preserves the Rundle LF, regarded as Late Eocene in age (Henstridge & Missen, 1982). Crocodiles in the Riversleigh LF’s range in age from Late Oligocene to modern (Archer, Hand & Godthelp, 1991), with Oligo–Miocene assemblages grouped as Faunal Zones A–D (Travouillon et al., 2006) recovered from freshwater carbonates referred to respectively as Depositional Phases 1–4. Faunal Zone A is interpreted to be Late Oligocene in age, Zone B as Early Miocene, Zone C as Middle Miocene and Zone D as possibly early Late Miocene (Woodhead et al., 2016). Northern Territory localities include Bullock Creek, on Camfield Station, and Alcoota Station, near Alice Springs, located in the northwest and south of the territory respectively. The Bullock Creek LF is regarded as a Middle Miocene contemporary of Riversleigh’s Faunal Zone C (Black et al., 2012) while the Alcoota LF is regarded as Late Miocene in age (Megirian et al., 2010). Institutional abbreviations: QM, Queensland Museum; NMV, Museum Victoria; NTM, Northern Territory Museum; AM, Australian Museum. Anatomical terms, illustrated in Fig. 2, follow Pol et al. (2012).

Figure 2 The modern crocodilian pelvic girdle.

Pelvic girdle of Crocodylus porosus (Schneider, 1801), (AM R131835) illustrating terminology used in this study. 1st sac. vert., first sacral vertebrae; 2nd sac. vert., second sacral vertebrae; ace., acetabulum; ace. perf., acetabular perforation; il., ilium; il. cr., iliac crest; il. pub. ped., pubic peduncle of the ilium; il. is. ped., ischial peduncle of the ilium; is., ischium; is. ant. ped., anterior peduncle of the ischium; is. dist. bl., distal blade of the ischium; is. ped. br., peduncle bridge of the ischium; is. post. ped., posterior peduncle of the ishium; M. il. caud., attachment site for the musculus ilioischio caudalis; M. pub. tib., attachment for the musculus puboischio tibialis; postace. pro., postacetabular process; preace. pro., preacetabular process; pub., pubi. Scale bar equals five centimetres.

To place the mekosuchine pelvic girdle into a phylogenetic context, iliac materials were scored according to character states modified in part from Brochu (1999), detailed in Appendix S1. One specimen, QM F57910, was excluded as the majority of characters could not be scored. Parsimony analysis of the resulting matrix (Appendix S2) was undertaken in PAUP 4.0 (Swofford, 2003). Maximum parsimony topographies were generated by the two step method used by Worthy et al. (2006) and performing 1,000 replicates at the initial step. As the most basal eusuchian taxon available for scoring, Gavialis gangeticus (Gmelin, 1789) was used as the outgroup to the Crocodylidae in which the Mekosuchinae currently nest (Brochu, 2003). The integrity of the selected character matrix was further tested using a bootstrap analysis of 1,000 replicates. Nexus format files of the data are provided as Supplemental Information.

Tingamarra, Queensland. QM F57908, right ilium; QM F57909, right ilium; QM F57910, left ilium; QM F54537, right ischium; QM F54533, right coracoid; QM F23586, left coracoid.

The Narrows Graben, Queensland. NMV P228164, right ilium; NMV P228627, right ilium; NMV P228635, left ilium; NMV P252361, left ilium; NMV P228615, right pubis; NMV P228623, right scapula; NMV P252360, right scapula; NMV P227038, left scapula; NMV P227803, left scapula; NMV P252359, right scapula and partial coracoid.

Riversleigh World Heritage Area, Queensland. Zone A. Quantum Leap Site: QM F41198, left ilium. Zone B. Price is Right Site: QM F57913, partial pelvis in articulation; QM F57914, first sacral vertebrae; QM F57912, right coracoid. Zone C. Golden Steph Site: QM F31406, right ilium; Ringtail Site: QM F40581, right ilium; QM F57911, right ilium.

Bullock Creek, Northern Territory. NTM P891-5, left ilium; NTM P908-35, left ilium; NTM P8695-272, right ischium; NTM P906-23, right ischium; NTM P895-139, right coracoid.

Alcoota Station, Northern Territory. NTM P5895, left ilium.

Results

Parsimony analysis

A total of 282 trees of 18 steps were generated. Specimens consistently organise into three clades, with a fourth in basal polytomy with the Gavialis outgroup (Fig. 3A), reflecting apparent generic or specific grouping. For the rest of this paper the term pelvic form is used to refer to these groups and the morphology of ilia and associated materials is subsequently discussed within this framework. These pelvic forms also display a consistent placement, pelvic forms three and four grouping as the most derived and form two placing basal to these. There is strong bootstrap support for the pelvic forms (Fig. 3B), but the relationships between forms is supported only insofar as forms three and four group more closely with respect to forms one and two.

Figure 3 Results of parsimony analysis.

Majority rule consensus trees resulting from (A), parsimony analysis of selected pelvic characters, (B), bootstrap analyses of selected pelvic characters. Trees are rooted with the Gavialis outgroup and pelvic forms are highlighted. Illustrated in TreeView (Page, 1996).

Pelvic form one

Right ilium QM F57908 (Figs. 4A–4C), right ilium QM F57909 (Fig. 4D), left ilium QM F57910 (Fig. 4E), and right ischium QM F54537 (Figs. 4F–4H). The postacetabular process of the ilium is robust, dorsoventrally taller than the short ischial and pubic peduncles and bluntly terminating in the posteriorly orientated articular surface of the M. ilioischio caudalis. The iliac crest expands dorsally, 30–40° from the sagittal plane. The articular surface of the pubic peduncle is orientated anteroventrally. The margin of the acetabular perforation between ischial and pubic peduncles is small and well defined. The acetabulum is shallowly concave. Attachment sites for the transverse processes of the sacral vertebrae are situated centrally on the ilium’s medial surface.

Figure 4 Pelvic form one.

QM F57908 in (A), lateral (B), medial and (C), posterior views, QM F57909 in (D), lateral view, QM F57910 in (E), lateral view, QM F54537 in (F), lateral (G), medial and (H), anterior views. All scale bars equal two centimetres.

The ischium is similarly robust. The preserved extent of the distal blade displays very little medial curvature. The posterior extension of the ischial arch is pronounced, reachingnearly twice the length of the proximal condyles. The anterior extent of the blade forms a thick, rugose margin. The muscle scar of the M. puboischio tibialis on the lateroanterior surface of the blade (Fig. 4F) is well developed.

Pelvic form two

Right ilium NMV P228164 (Figs. 5A–5C), left ilium NMV P228635 (Fig. 5D), left ilium, NMV P252361 (Fig. 5E), right ilium NMV P228627 (Fig. 5F), right ilium QM F40581 (Fig. 5G), and right pubis NMV P228615 (Fig. 5H). The postacetabular process of the ilium is larger compared to pelvic form one, both anteroposteriorly elongate and dorsoventrally deep. The iliac crest orientates 30–40° from the sagittal plane similar to pelvic form one and forms a prominent convex arch just posterior to the ischial peduncle. Consequently, the preacetabular process is situated between concave margins on the iliac crest. The attachment for the M. ilioischio caudalis forms a posteriorly extending terminal knob distinct from the margin of the iliac crest (Figs. 5A, 5D and 5E–5F). The pubic and ischial peduncles are more developed than those displayed by pelvic form one, proportionally deep even in relation to the expanded postacetabular process. Similar to pelvic form one, however, the acetabulum is shallowly concave, the pubic peduncle is anteroventrally orientated and the perforation between peduncles is well defined. Attachment scars for the sacral vertebrae are situated centrally on the medial surface of the ilium. QM F40581 is similar in profile but with a less developed terminal knob (Fig. 5G). The pubis is expanded mediolaterally, displaying a well-developed medial condyle and pronounced rugosities on the dorsolateral surface of the distal blade.

Figure 5 Pelvic form two.

NMV P228164 in A, lateral B, medial and C, dorsal views, NMV P228635 in D, lateral view, NMV P252361 in E, lateral view, NMV P228627 in F, lateral view, QM F40581 in G, lateral view, NMV P228615 in F, dorsal view, All scale bars equal two centimetres.

Pelvic form three

Left ilium QM F41198 (Figs. 6A–6C), right ilium QM F57911 (Fig. 6D), left ilium NTM P908-35 (Fig. 6E), left ilium NTM P891-5 (Fig. 6F), left ilium NTM P5895 (Fig. 6G), right ischium NTM P906-23 (Fig. 6H), and right ischium NTM P8695-272 (Fig. 6I). The preacetabular process is situated towards the anterior of the iliac crest, encroaching onto the pubic peduncle. The postacetabular process is gracile compared to pelvic forms one and two, dorsoventrally shallower than the deep ischial and pubic peduncles, and bluntly terminates in the posterodorsally orientated articular surface of the M. ilioischio caudalis. The iliac crest displays a slight lateral expansion, orientating 40–60° from the sagittal plane. The pubic peduncle expands anteriorly such that it is level with the ischial peduncle. The iliac portion of the acetabulum is nearly imperforate in consequence, the margin between peduncles not greatly penetrating into the acetabular space. The acetabulum is deeply concave. The attachments for the sacral vertebrae are situated centrally on the medial surface of the ilium.

Figure 6 Pelvic form three.

QM F41198 in A, lateral B, medial C, posterior views, QM F57911 in D, lateral view, NTM P908-35 in E, lateral view, NTM P891-5 in F, lateral view, NTM P5895 in G, lateral view, NTM P906-23 in H, lateral view, NTM P8695-272 in I, lateral view. All scale bars equal two centimetres.

The distal blade of the ishium (Figs. 6H and 6I) does not expand as posteriorly as that of pelvic form one, being closer in proportion to the columnar blade of modern crocodilians. The distal blade is also more gracile than in pelvic form one, lacking the thick anterior margin and prominent rugosity. Despite reduction of the iliac margin of the acetabular perforation, the ischial margin is well defined by the bridge of the ischial peduncles.

Pelvic form four

Partial left pelvis QM F57913 (Figs. 7A–7E) and right ilium QM F31406 (Fig. 7F). The postacetabular process is dorsoventrally deep and both iliac crest and postacetabular process expand laterally 60–70° from the sagittal plane, enclosing a deeply concave acetabulum dorsally (Fig. 7C). The articular surface for the M. ilioischio caudalis forms a posterodorsally facing rim, distinctly raised from the margin of the iliac crest (Figs. 7A and 7F). The dorsal rugosity of the iliac crest is well developed. Sites for the attachment of the transverse processes of the sacral vertebrae are situated ventrally on the medial surface of the ilium. The transverse process of the posterior sacral vertebrae is entirely excluded from the ventral margin of the postacetabular process and confined to the ischial peduncle. The attachment of the anterior sacral vertebrae has shifted anteroventrally with the preacetabular process, such that a distinct space now separates both attachment sites. As a consequence, the medial surface of the iliac crest is larger compared to the other pelvic forms and a large portion of the posterior iliac crest would have extended beyond the sacral vertebrae. Similar to pelvic form three, the ischial peduncle is proportionally deep compared to the shallow postacetabular process, with a mediolaterally wide and anteroposteriorly compressed articular face that forms a broad triangular surface which encloses the acetabulum posteriorly. The transverse process preserved on the first sacral vertebrae QM F57914 (Fig. 7G) is mediolaterally compact in proportion and anteroposteriorly compressed into a curved subquadrate lamina of bone displaying little posterior extension of the kind typically seen in modern crocodilians. This corresponds to the attachment sites displayed by pelvic form four.

Figure 7 Pelvic form four.

QM F57913 in A, lateral B, medial C posterior views, pubis in D, dorsal and E, lateral views, QM F31406 in F, lateral view, QM F57914 in G, lateral view. All scale bars equal two centimetres.

The ischium is similar in proportions to that of pelvic form one, displaying an anteroposteriorly expansive distal blade that is wider than the proximal head. The blade is gracile, however, similar to pelvic form three. The anterior peduncle is expanded, forming a broad ovoid ventral margin to the acetabular perforation. The head of the anterior peduncle is anteroposteriorly compressed, forming an enclosed anterior wall of the acetabulum. The posterior peduncle is similarly anteroposteriorly wide, with a mediolaterally expandedanterior margin that articulates with the posterior peduncle of the ilium to enclose the acetabulum posteriorly. The pubis is gracile, anteroposteriorly elongate and displays only slight ventral curvature in medial view.

Pectoral girdle material

Left coracoid QM F23586 (Fig. 8A), right coracoid QM F54533 (Fig. 8B), left scapula NMV P227803 (Fig. 8C), and right scapula NMV P252360 (Fig. 8D). The scapula and coracoid are moderately robust elements in these specimens, the distal blade of the coracoid being approximately twice the depth of the proximal head and the diaphysis greater than half the width of the proximal head and distal blade. The anterior process of the proximal head is anteroposteriorly short and blunt.

Figure 8 Pectoral girdle material.

QM F23586 in A, lateral view, QM F54533 in B, lateral view, NMV P227803 in C, lateral view, NMV P252360 in D, lateral view, NTM P895-139 in E, lateral view, NMV P228623 in F, lateral view, NMV P227028 in G, lateral view, NMV P252359 in H, anterior view, and QM F57912 in I, lateral view. All scale bars equal two centimetres.

Right coracoid NTM P895-139 (Fig. 8E). This specimen is a very robust element. The distal blade is short compared to the other specimens, about equal to the proximal head in dorsoventral depth, and separated by a robust bridge about half the length of the proximal head. The anterior process is short and blunt.

Left scapula NMV P227038 (Fig. 8F), right scapula NMV P228623 (Fig. 8G), right scapula and partial coracoid NMV P252359 (Fig. 8H), and right coracoid QM F57912 (Fig. 8I). Both scapula and coracoid are elongate and gracile elements in these specimens. The distal blade of the scapula is approximately two and a half times the length of the distal head and the diaphysis is less than half the width of the distal blade in both coracoid and scapula. The articular surface of the glenoid fossa in the coracoid is anteroposteriorly wide, more so than the glenoid displayed by the other specimens. Both scapula and coracoid display expansive and acute anterior processes of the glenoid joint.

Discussion

Fossils described here afford an opportunity to examine changes in the mekosuchine pelvic girdle during the early to middle Cenozoic. Pelvic form one appears to be representative of a species of Kambara, having been collected in association with skulls referable to that genus at Tingamarra. This includes QM F57910, suggesting this specimen pertains to pelvic form one rather than two, despite its fragmentary state. These Early Eocene fossils represent the oldest mekosuchine materials known and potentially shed light on what could be plesiomorphic features in mekosuchines. In form one, the ilium is robust, resembling gavialids and alligatorids in general morphology, with proportionally shallow ischial and pubic peduncles, shallowly concave acetabulum, and rounded postacetabular process that displays little of the dorsal constriction typical of crown group crocodylids (Figs. 2, 4A and 4D) (Brochu, 1999). This is notable in itself, suggesting an earlier radiation of mekosuchines within the Crocodylia than indicated by phylogenies based on crania, which place Mekosuchinae within the Crocodylidae (Salisbury & Willis, 1996; Brochu, 2003). In most other respects the ilium resembles the typical eusuchian system, displaying an acetabulum and iliac crest orientated in the sagittal plane and sacral articulation centrally situated on the medial face (Figs. 4B–4C) (Brochu, 1999).

Significant differences in the acetabulum and iliac crest can be seen in pelvic form two, known from the Late Eocene, and shared with Oligo–Miocene pelvic forms three and four. The acetabulum in these forms is characterised by increasing peduncle depth and, in the Oligo–Miocene forms, concavity. In all Oligo–Miocene forms, the ischial peduncle increases in proportional depth to the iliac crest, deepening the pelvis overall (Figs. 5A, 6A and 7A). Pelvic form three (Figs. 6A and 6D–6F) displays anteroventral expansion of the pubic peduncle, which reduces the extent of the iliac portion of the acetabular perforation. The pelvic girdle remained perforate in mekosuchines, evidenced by the ischia described here (Figs. 6H–6I and 7A). The anterior extent of pelvic form four is unknown but marked anterior expansion of the pubic peduncle in the ischium suggests a similar morphology to pelvic form three (Fig. 7A). Both pelvic forms three and four display increasing concavity of the acetabular space (Figs. 6A, 6D–6G and 7A). Pelvic form two, by contrast, does not display expansion of the pubic peduncle and remains only slightly concave in Oligo–Miocene taxa (Figs. 5A and 5G).

This same division between pelvic form two and forms three and four is apparent in the changes of the iliac crest, characterised by development of a distinct terminal knob on the postacetabular process, evidently convergent on but not identical with the constriction seen in crown group crocodylids (Brochu, 1999), and expansion of the crest itself. Unlike the crocodylid constriction, the terminal knob in pelvic form two appears to develop by a posteriorly orientated expansion of the iliac crest (Figs. 5A and 5D–5F). In pelvic form three the expansion has become pronounced enough to subsume the postacetabular process, but in contrast to pelvic form two is now posterodorsally orientated (Figs. 6A and 6E), and occurs as a dorsal rim in pelvic form four (Fig. 7F). Pelvic form two displays a dorsally expansive iliac crest similar to pelvic form one (Figs. 5A and 5D). The crest of pelvic form three is somewhat ambiguous, displaying slight lateral expansion (Fig. 6C) but the same sacral articulation as forms one and two. The crest is greatly modified in pelvic form four, enclosing the acetabulum dorsally, while the laterally expansive pubic and ischial peduncles enclose the acetabulum anteroposteriorly (Fig. 7C). Pelvic form three is associable with crania of Baru darrowi (Willis, Murray & Megirian, 1990) in the Bullock Creek LF. The highly modified morphology of the ilium in pelvic form four suggests affinity with the similarly highly derived Quinkana meboldi (Willis, 1997b) of Riversleigh’s Miocene faunas, but associated materials will be necessary to confirm this. Ontogeny is another possible explanation for the close resemblance between pelvic forms three and four, although dissimilar sacral articulation and iliac crest development across a range of sizes suggest they represent different taxa rather than different developmental stages within one species.

These different morphologies of the pelvic forms, together with their relative ages, suggest separate successive radiation events corresponding to the climatic optima of the Eocene and Miocene, a common trend observed in many early Cenozoic crocodilian communities (Brochu, 2003; Mannion et al., 2015). Buchanan (2009) observed that the well diversified Miocene radiation of mekosuchines is likely to have been derived from an Eocene radiation restricted to the Australian continent. It is possible pelvic form two represents a separate species of Kambara, with diversification in the genus more conspicuous in the postcrania than crania. However, pelvic form two persists into Oligo–Miocene Riversleigh faunas, from which Kambara has not been reported. It is more likely that pelvic form two represents a different genus, suggesting greater generic diversity in the Eocene than is reflected in known skull forms. The Oligo–Miocene presence of pelvic form two, coupled with the fact that none of NMV P228164, NMV P228635, NMV P252361, NMV P228627 or QM F40581 are greater than five centimetres in antero-posterior length, suggests it may represent Mekosuchus, a genus of suspected dwarf species (Willis, 1997b), although this remains speculation until skeletal association with the cranium can be made. If such is the case, these ilia would push the origins of Mekosuchus and tribe Mekosuchini earlier into the Eocene.

These interpretations are supported by the results of the parsimony analysis that generally places pelvic forms one and two basal to forms three and four, in agreement with their relative ages (Fig. 3A). Because the mekosuchine pelvis cannot be distinguished on the basis of a strict apomorphy, and appears to have developed the constriction of the postacetabular process of crown group crocodylids convergently, any parsimony analysis is likely confounded by some degree of homoplasy. The effect of this can been seen in the loss of tree resolution under bootstrap analysis, although the later Oligo–Miocene forms remain distinct (Fig. 3B). Coupled with the more gavialid/alligatorid features displayed by pelvic form one, this warrants revaluation of the base relationships of the Mekosuchinae within the Eusuchia. While this falls beyond the scope of the present paper, work in preparation by one of us (AY) aims to revaluate mekosuchine taxonomic relationships within a more robust phylogenetic framework than previously available.

The lateral expansion of the iliac crest, accompanied by ventral migration of the transverse processes of the sacral vertebrae in pelvic form four is the most conspicuous development of the mekosuchine pelvis, with substantial effect (Figs. 7B–7C). First, this arrangement increases the medial and lateral surface area of the iliac crest. Second, it re-orientates the acetabulum ventrolaterally, with much of the dorsal extent of the acetabulum effectively facing ventrally. Third, the postacetabular process extends further posterior to the sacral vertebrae compared to the situation in modern crocodilians. This highly derived morphology is remarkable in its similarity to the girdle reported in sebecosuchians (Colbert, 1946). The iliac crest of the terrestrial South American Eocene Sebecus icaeorhinus (Simpson, 1937) displays similar lateral orientation and expansion and ventral migration of the transverse processes of the sacral vertebrae (Pol et al., 2012). When mekosuchines were first described, a possible relationship with Paleogene sebecosuchians was suggested (Hecht & Archer, 1977; Megirian, 1994). Willis (1993) established Mekosuchinae as a subfamily within Eusuchia and attributed their similarity to sebosuchians to evolutionary convergence. This appears also to be the case for postcrania. The earliest (and perhaps least derived) form of the mekosuchine ilium bears little resemblance to that seen in the contemporaneous S. icaeorhinus.

This convergent morphology suggests pelvic form four was similarly adapted for pillar-erect stance, as Pol et al. (2012) infers occurred in S. icaeorhinus. Erect pelvic stance has evolved independently several times in amniotes by way of two systems (Benton & Clark, 1988; Schachner, Manning & Dodson, 2011). The first, buttress-erect case characterises the bird-line archosaurs and mammals. The proximal femur has a medial head or ball which articulates with a defined socket on the pelvic girdle, and lateral trochanters allowing the limb to operate in a sagittal arc of motion while maintaining an effective line of action for the pelvic musculature by its insertion onto the trochanter. The second, pillar-erect case characterises the bulk of early archosaurs and Mesozoic crocodylomorphs (Parrish, 1987; Sereno, 1991). Instead of the femur being modified, the pelvis is laterally expanded. This results in articulation of the limb in a ventral position, again allowing a sagittal arc of motion while maintaining an effective line of action for the appendicular musculature from their origins now situated over the limb.

Pelvic form four appears to show modifications towards the pillar-erect case. The lateral reorientation of the iliac crest shifts the origins for much of the dorsal medial musculature of the ilium dorsolaterally with respect to the proximal head of the femur. Affected musculature includes many of the main abductors and flexors of the hind-limb involved during the stance phase of the step cycle. These include the M. ilio-femoralis, M. ilio-tibialis, M. ilio-fibularis, M. femero-tibialis externus, the dorsal branch of the M. femero-tibialis internus and the M. ilio-costalis (Romer, 1923a; Romer, 1923b; Hutchinson & Gatesy, 2000). The effect would be to increase the lever arm of these muscles to the femur when held near the sagittal plane, similar to the effect of the greater trochanter in erect stance in mammals (Benton & Clark, 1988).

This has consequences for postural range, however. Among eusuchians, various postures are adopted during locomotion, ranging from lateral sprawling to an erect high-walk and gallop (Benton & Clark, 1988; Reilly & Elias, 1998; Hutchinson & Gatesy, 2000). Kinematics between postures are identical, postural grades differing only in the orientation of the femur (Reilly & Elias, 1998). Freedom of orientation is enabled by the open planar surface of the acetabulum and accompanying lateral muscular origins on the iliac crest, which afford a significant line of action to the limb musculature across a wide arc (Reilly & Elias, 1998). Pelvic forms one and two are similar to this basal eusuchian system and hence these probably operated in a similar way.

Conversely the acetabulum is a more confined space in pelvic forms three and four: in the former by expansion and increasing concavity, and in the latter by enclosure of the acetabulum dorsally and ventrolaterally. The lateroventral acetabulum in pelvic form four results in the origins of the femoral abducting musculature being orientated ventrally, particularly the M. ilio-femoralis and M. ilio-tibialis (Romer, 1923a; Hutchinson & Gatesy, 2000). While the lever arm of this musculature would be increased when the femur is held close to the sagittal plane, it would be similarly decreased when held towards the coronal plane. This suggests that the capacity for sprawling gait was diminished in pelvic form four.

Fundamental to these modifications, however, is the expansion of associated ventral elements of the pelvic girdle in the Early Eocene species. This extends the lever arms of muscle groups involved in both the stance and swing phases and support of the body against abduction induced by gravity. The posterior expansion of the ischial arch extends the M. ischio trochantericus, branches of the M. adductor femoris, and third branch of the M. pubo-ischio femoralis externus. Expansion of the pubis extends the first and second branches of the M. pubo-ischio femoralis externus (Romer, 1923a; Romer, 1923b). The extent of the rugosity on the pubis in pelvic form two indicates marked development of the second branch. Ventral migration of the transverse processes on the ilium similarly extends the medial musculature of the ilium posteriorly, and excludes the pubo-ischio femeralis internus from the posterior medioventral surface of the ilium (Romer, 1923a; Romer, 1923b). It is possible that origin of this muscle on the ilium was eliminated entirely or at least much of its posterior extent reduced. It is also possible that it migrated ventrally as well onto the anteroposteriorly expansive bridge between the peduncles of the ischium. The exception is the ischium of pelvic form three. Despite close morphological similarity with the apparently derived pelvic form four, pelvic form three’s ventral musculature would appear to have converged on that of crown group crocodylids. This further supports a second radiation event, but one with a morphological fuse that extended back to an earlier Eocene radiation, as the source of the Oligo–Miocene pelvic forms.

Similar morphological diversification in the pectoral girdle appears concomitant with that of the pelvic girdle. Eocene fossils, likely pertaining to Kambara (Figs. 8A–8B), may again represent the plesiomorphic condition for mekosuchines. The proportions are very similar to those found in modern crocodilians. By the Oligo–Miocene, both highly gracile and robust elements had developed, resulting in a similar division between girdles that retained more typically crocodilian proportions (Fig. 8E) and those that display a distinct alteration in the muscle origins of the articulating limb (Figs. 8F–8I). The proportionally longer distal blade of the scapula in NMV P227038, NMV P228623 and NMV P252359 (Figs. 8F–8H), results in an increase in cross-sectional area and length of the lever arm which would be important in the main stance phase involving the stabilising musculature of the humerus. These muscles would include the M. teres major, M. deltoideus scapularis and M. subscapularis (Meers, 2003). The M. coracobrachialis brevis ventralis should be included among these if the distal blade of the coracoid was similarly lengthened. In contrast with the pelvic girdle, however, the evolution of the articular surface of the pectoral girdle appears to be more conservative. The glenoid is directed at the same angle to the proximal head in the coronal plane in later Oligo–Miocene materials, the only variation being the somewhat wider glenoid facet on the gracile coracoid QM F57912 (Fig. 8I). This potentially allowed a greater range of motion of the humerus but this is difficult to tell without the proximal humeral head to enable detailed analysis of the articular surfaces.

Mekosuchines appear therefore to have undergone a diversification of locomotory strategy by the Oligo–Miocene. This has interesting palaeoecological implications regarding mekosuchine faunas of the Riversleigh World Heritage Area. Pelvic forms two, three and four are all found in possibly close temporal proximity at Golden Steph and Ringtail Sites in Riversleigh’s Faunal Zone C assemblage. Ringtail Site notably preserves a diverse set of mekosuchine species in apparent sympatry (Willis, 1993; Willis, 2001). A distinct division of cranial shapes is also present allowing for the niche specialisation of prey (Willis, 2001). Locomotory diversification suggests another dimension of specialisation that complements cranial divisions, potentially allowing the same prey divisions to be pursued in both semi-aquatic and fully terrestrial hunting ranges.

Conclusions

Diversification of cranial morphology in mekosuchine crocodiles between the Eocene and Oligo–Miocene was matched by similar diversity in their pectoral and pelvic girdles. Assuming that Eocene Kambara species exhibit the plesiomorphic state of the mekosuchine pelvis, it is similar to that in extant gavialids and alligatorids. More specialised forms similar to crown group crocodylids apparently developed secondarily by the Oligo–Miocene. One lineage developed a progressively enclosed acetabulum and laterally expanded iliac crest. With elongation of elements of the pubis and ischium, the line of action of the femur in these pelves shifted ventrally towards the sagittal plane. This notably resembles the organisation in pillar-erect crocodylomorphs of the Mesozoic. Another lineage, by contrast, seems to have retained the structure of the pelvis seen in modern crocodilians, suggesting that members of this lineage would have exhibited a variable gait. There therefore appears to have been a diversification of locomotory strategies in mekosuchine crocodiles during this time period in tandem with diversification of cranial shape, suggesting this is an important dimension in the process of speciation in crocodilians. If the pillar-erect lineage is referable to Quinkana it would be indicative of a greater focus on cursorial movement, and the terrestrial sphere, in agreement with the derived features of the quinkanine cranium.

Supplemental Information

Appendix S1 Selected pelvic characters

Characters used in parsimony and bootstrap analysis of the pelvic materials. Characters 2,6, 8 are partially modified from characters 28 and 34 of Brochu (1999).

Click here for additional data file.

Appendix S2 Character-taxon matrix used for parsimony analysis

Characters scored according to the criteria detailed in Appendix S1 for each specimen and the pelvis of Gavialis gangeticus. Characters were unweighted and multistate characters were left unordered.

Click here for additional data file.

Supplemental Information 1 Character-taxon matrix used for parsimony analysis

Characters scored according to the criteria detailed in Appendix S1 for each specimen and the pelvis of Gavialis gangeticus. Characters were unweighted and multistate characters were left unordered.

Click here for additional data file.

We thank S Hocknull of the Queensland Museum, Brisbane, E Fitzgerald of Museum Victoria, Melbourne and J Scanlon, of the Riversleigh Interpretive Centre, Mount Isa for access to comparative materials. We thank A Gillespie for expert preparation of the specimens described here. We also thank the editor and reviewers for their constructive criticisms of drafts of the manuscript.

Additional Information and Declarations

Competing Interests

Author Contributions

Data Availability

The authors declare there are no competing interests.

Michael D. Stein conceived and designed the experiments, performed the experiments, analyzed the data, wrote the paper, prepared figures and/or tables, reviewed drafts of the paper.

Adam Yates analyzed the data, contributed reagents/materials/analysis tools, reviewed drafts of the paper.

Suzanne J. Hand and Michael Archer conceived and designed the experiments, analyzed the data, reviewed drafts of the paper.

The following information was supplied regarding data availability:

The raw data has been supplied as a Supplementary file.

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
