# Peer review of "Variation in the pelvic and pectoral girdles of Australian Oligo–Miocene mekosuchine crocodiles with implications for locomotion and habitus"

_PeerJ, doi:10.7717/peerj.3501_

## Round 0.1 · original submission · Major Revisions

· Academic Editor

Major Revisions

The reviewers agree that there are concerns about the polytomy in the phylogeny that makes inferring the ancestral/basal state for the clade entirely ambiguous and thus some conclusions untenable. Rather than remove the phylogenetic components from the study, which make it truly comparative and potentially valuable, this needs to be addressed for the study to be acceptable. And this cannot be addressed by choosing a phylogeny that gives the desired results, so this is a challenging problem. Reviewer 1 also raises issues about the pelvic morphologies assumed as "forms". Please make revisions and include a Response addressing all reviewers' points if you resubmit. The paper will need re-review to ensure that the reviewers (and I) are convinced.

·

Basic reporting

The manuscript is clearly written, well cited and discussed.

Experimental design

I would like more information in regards to your character matrix, particularly with regards to the Brochu 1999 paper. I was only able to align two of your characters with that matrix (being 3 and 6 which align to number 34 and 28 respectively in the Brochu list). Can you clarify your other characters?

I also worry that you are pre-emptively grouping your specimens into forms 1-4 for the pelvis might lead to wrong results or over clumping of specimens (see comments on validity). Ideally you would be running the analysis on each of the bones without the a priori assumptions and putting in ? for the missing character states. If you did do this, this needs more clear explanation in the methods and maybe some trees to show this in the supplementary information. A similar issue arises with the shoulder (forms 1-3) which was not subjected to any phylogenetic testing.

Validity of the findings

It is difficult to say how accurate these findings are due to what appears to be the a priori assumptions that the bones all fit into forms 1-4 for the pelvis. It would be ideal if the analysis was re-run without the assumption, and then see if the same forms fall out as clades. Based on their extensive descriptions of the forms and the traits linking them, I would suggest they should. Additionally, if the forms hold up to additional testing, it is interesting that you find 3 pelvic forms in the Golden Steph site alone which bears further discussion (all based on ilia).

Finally, based on Figure 8 there is an issue with the phylogenetic results and discussion. Figure 8, as displayed now, shows a single basal polytomy uniting all forms with the outgroups. As such Forms 1 and 2 are not more basal and 3 and 4 are not more derived (although 3 and 4 are more closely related to each other).

Reviewer 2 ·

Basic reporting

The paper is very well written and clearly presented. Although I am not an expert on pelvic girdle anatomy in mekosuchine crocodiles, the anatomical descriptions seem thorough, and are well integrated with high quality figures (Figs 1-7).

Experimental design

The paper works well as an anatomical overview of girdle evolution in mekosuchines during the Eocene to Middle Miocene. However, I do have major concerns about the phylogenetic analysis presented and the interpretation of the results. The results from the phylogenetic analysis are presented in Figure 8, which although labelled as a 50% majority rule tree, actually seems to represent both the 50% majority rule tree and the strict consensus tree (the only branch that has any phylogenetic meaning has 100% support). In the text, the authors claim that:

"Pelvic forms one and two are placed basal to the derived pelvic
forms three and four under both strict and majority rule consensus (Fig 8)." *Lines 240,241

and

"These interpretations are congruent with the results of the parsimony analysis that place pelvic forms one and two consistently basal to forms three and four (Fig. 8)." *Lines 314,315

When viewing the tree presented in Fig.8, this does not appear to be the case. Instead, what is presented is a large polytomy, where there is no indication that Forms 1 and 2 are more basal than Forms 3 and 4 in the concensus tree. The only phylogenetic information presented in Fig.8 is that Forms 3 and 4 are more closely related to each other than they are to other morphotypes. There is no indication in the presented figure of what forms are basal with respect to the outgroup(s). With the exception of the relationship between Forms 3 and 4, the topology is a star tree. This suggests that either the figure is not drawn correctly, the wrong figure has been uploaded, or the authors have misinterpreted the results. The inconclusive phylogenetic results likely results from a low number of 11 characters - which should be reported in the text. I recommend that the authors either:

1. Reinterpret their phylogenetic results and modify all references to it in the text
2. Add more characters to try and improve the resolution of the tree
3. Remove the phylogenetic analysis and focus on the detailed anatomical descriptions and narrative.

Validity of the findings

As discussed above, modifications are needed to the phylogenetic analysis presented in the paper, as currently the interpretation of the presented consensus topology is incorrect. Any conclusions based on the phylogenetic analyses are not sound.

---

## Round 0.2 · Minor Revisions

· Academic Editor

Minor Revisions

I agree with the reviewers that the MS is improved substantially and there are only minor tweaks left to be made. Please do consider the rooting issue raised. In your Response with the revised MS please be sure to address all points. I hope to be able to accept the MS then; I do not expect further review to be needed.

·

Basic reporting

No comment

Experimental design

No comment

Validity of the findings

No comment

Additional comments

Having re-reviewed the manuscript I believe there are no major corrections remaining.

Minor points:
Ln147: I remain uncomfortable calling Forms 3 + 4 derived with respect to 1 + 2 when they all share the basal polytomy (Fig 3B) with the outgroup.

Ln 342: Sebecus icaeorhinus Simpson, 1937. Think there should be brackets around Simpson, 1937.

I would suggest the authors check the formatting of the phylogenetic figure (3) as I cannot read any of the specimen numbers (I presume that is what is there).

Reviewer 2 ·

Basic reporting

The standard of presentation is generally excellent and the paper is well cited. I only recommend a few minor additions/modifications in this revised version:

Line 49 in the introduction, when discussing evidence of morphological diversification in fossil crocodylomorphs. You could cite the following studies exploring this morphological diversification numerically:

Wilberg, E. W. (2017). Investigating patterns of crocodyliform cranial disparity through the Mesozoic and Cenozoic. Zoological Journal of the Linnean Society. https://doi.org/10.1093/zoolinnean/zlw027

Stubbs, T. L., S. E. Pierce, E. J. Rayfield, and P. S. L Anderson. (2013) Morphological and biomechanical disparity of crocodile-line archosaurs following the end-Triassic extinction. Proc. R. Soc. B, vol. 280, no. 1770, p. 20131940. DOI: 10.1098/rspb.2013.1940

Line 114 " Minimum branch-length topographies were generated"
I presume this simply means the most parsimonious trees were searched for, or the trees with fewest steps were searched for? Using the term 'minimum branch lengths' could suggest some type of time-scaling - where individual branches in the tree have variable lengths (in geological time). I would maybe rephrase this to avoid such connotations.

Finally, you could increase the text size for the labels in the phylogeny figure - the labels are currently quite difficult to read.

Experimental design

The modifications the authors have made now resolve the methodological issues previously identified.

Validity of the findings

I ran the authors data matrix in PAUP using their phylogenetic protocol, and I recovered the same result - 282 trees with lengths of 18 steps. I am satisfied that the modifications they have made to their analysis, the results they obtained and the discussion of their results.

One note, is the tree the authors present rooted? One reason the authors recover a basal polytomy with the outgroup and Pelvic Form One could be because the tree is not rooted. This is not a fault, just a thought.

---

## Round 0.3 · accepted · Accept

· Academic Editor

Accept

I am satisfied with the revisions. The polytomy and rooting issue still concern me; the ancestral state for the clade remains very uncertain; but I think this is something that future work can resolve, and this may well depend (as the paper states) on where mekosuchines fit into the broader phylogeny, which is beyond this paper's scope. This aside, the paper is interesting and has been thoroughly reviewed and thus I see plenty of good reasons to publish it now. Congratulations!